# Regulation of Immune Checkpoint Antigen CD276 (B7-H3) on Human Placenta-Derived Mesenchymal Stromal Cells in GMP-Compliant Cell Culture Media

**DOI:** 10.3390/ijms242216422

**Published:** 2023-11-16

**Authors:** Bastian Amend, Lea Buttgereit, Tanja Abruzzese, Niklas Harland, Harald Abele, Peter Jakubowski, Arnulf Stenzl, Raphael Gorodetsky, Wilhelm K. Aicher

**Affiliations:** 1Department of Urology, University Hospital, Eberhard Karls University, 72076 Tuebingen, Germany; 2Centre for Medical Research, Department of Urology, Eberhard Karls University, 72076 Tuebingen, Germany; 3Department of Gynaecology and Obstetrics, University Hospital, Eberhard Karls University, 72076 Tuebingen, Germany; 4Biotechnology and Radiobiology Laboratory, Sharett Institute of Oncology, Hadassah-Hebrew University Medical Centre, Jerusalem 91120, Israel

**Keywords:** maternal and fetal placenta mesenchymal stromal cells, immune checkpoint antigen, CD276, B7-H3, cell therapy

## Abstract

Therapies utilizing autologous mesenchymal cell delivery are being investigated as anti-inflammatory and regenerative treatments for a broad spectrum of age-related diseases, as well as various chronic and acute pathological conditions. Easily available allogeneic full-term human placenta mesenchymal stromal cells (pMSCs) were used as a potential pro-regenerative, cell-based therapy in degenerative diseases, which could be applied also to elderly individuals. To explore the potential of allogeneic pMSCs transplantation for pro-regenerative applications, such cells were isolated from five different term-placentas, obtained from the dissected maternal, endometrial (mpMSCs), and fetal chorion tissues (fpMSCs), respectively. The proliferation rate of the cells in the culture, as well as their shape, in vitro differentiation potential, and the expression of mesenchymal lineage and stem cell markers, were investigated. Moreover, we studied the expression of immune checkpoint antigen CD276 as a possible modulation of the rejection of transplanted non-HLA-matched homologous or even xeno-transplanted pMSCs. The expression of the cell surface markers was also explored in parallel in the cryosections of the relevant intact placenta tissue samples. The expansion of pMSCs in a clinical-grade medium complemented with 5% human platelet lysate and 5% human serum induced a significant expression of CD276 when compared to mpMSCs expanded in a commercial medium. We suggest that the expansion of mpMSCs, especially in a medium containing platelet lysate, elevated the expression of the immune-regulatory cell surface marker CD276. This may contribute to the immune tolerance towards allogeneic pMSC transplantations in clinical situations and even in xenogenic animal models of human diseases. The endurance of the injected comparably young human-term pMSCs may promote prolonged effects in clinical applications employing non-HLA-matched allogeneic cell therapy for various degenerative disorders, especially in aged adults.

## 1. Introduction

Therapies based on mesenchymal stromal cells (MSCs) have been widely tested and employed in numerous preclinical studies and clinical applications [1,2,3,4,5,6,7,8,9,10]. The main source of MSCs has been bone marrow aspirates, where a small fraction of bone marrow-derived MSCs (bmMSCs) can be isolated, separated from the hematopoietic progenitors, and further expanded in cell cultures [3,11]. The expression of CD73 (5′-ectonucleotidase), cell surface molecule CD90 (Thy-1), and CD105 (Endoglin, involved in TGF-β-signaling) were defined as consensus-positive cell surface markers for bmMSCs [12,13]. These antigens are also present in various other types of MSCs, including adipose tissue-derived (atMSCs) [14,15] and placenta-derived (pMSCs) cells [10,16,17,18,19,20,21,22]. MSCs have often been called “stem cells” because of their differentiation capabilities. However, in practice, their impact in many cases has been observed to be indirect, rather than serving as the primary building blocks for differentiated tissues [4,10,21,23,24]. A drawback in bmMSC-based therapies could be associated with undesired complications, such as spontaneous heterotopic mineralization [25,26]. The promising results observed following different bmMSC transplantations, however, were also found to have a limited yield and availability, instigating the search for similar and easily available cells in other tissues and organs [14,27,28,29,30,31,32].

Autologous cells are the preferred choice for cell-based therapies as they reduce the risks of their fast rejection and in the prevention of disease transmission [33,34,35,36,37]. The primary potential drawback in using autologous cells for pro-regenerative treatments in aged individuals is their anticipated sub-optimal expansion efficacy and lower regenerative effectiveness compared to cells derived from younger individuals [38,39,40]. This seems to be in line with the observations of the better pro-regenerative activity of MSCs from young donors compared to MSCs from aged ones [41], which is also manifested in the secretion of factors associated with wound healing and neo-vascularization [42].

In the attempt to isolate autologous MSCs for cell therapies, different techniques for cell enrichment have been examined [43,44,45]. Moreover, different cell sources have been investigated. A commonly tested alternative for bmMSCs was atMSCs [14]. However, a major drawback of MSCs isolated from senior patients for autologous therapies may be the relative senescence of such cells, likely reducing their regenerative effectiveness upon implantation [38,39,46,47]. Overall, MSCs are tolerated comparably well following even allogeneic or xenogeneic transplantation due to their low expression of HLA class I and HLA-DR antigens [12,23,48].

Human placenta tissues may be an easily available source for the fast and simple isolation of young, HLA-low, and regeneration-competent MSCs [22,49,50,51,52,53,54,55]. Despite their allogeneic or xenogeneic origin, these cells were shown to induce a very significant indirect pro-regenerative effect following their transplantation, as has been demonstrated in clinical studies and numerous preclinical animal models [21,22,55,56,57,58,59,60]. The mpMSCs can be isolated in very high numbers from the well-vascularized placenta tissue of relatively young mothers. The fpMSCs are even younger cells of the embryo, and their average telomeres were shown to be, apparently, rather long [61], which may allow the expansion of minor cell subsets in vitro with numerous cell passages before reaching replicative senescence [18]. The pMSCs seem to maintain their stable phenotype and, therefore, are much less prone to spontaneous osteogenic differentiation compared to bmMSCs [17,18,19,26].

Elevated expression of immune-checkpoint cell surface antigen CD276 on tumor cells has been reported to be associated with poor prognosis, suggesting the suppression of anti-tumor immune responses of host T lymphocytes towards the CD276-bearing tumor cells [62,63,64,65]. Wharton’s jelly-derived MSCs (wjMSCs), which are assumed to have a similar phenotype to fpMSCs, were also shown to express CD276 in situ and in vitro [54,66]. Based on these observations, it is expected that the low expression of HLA class I and II molecules, in conjunction with the increased expression of the immune checkpoint antigen CD276 on pMSCs, may potentially downregulate allogeneic T-lymphocyte responses, potentially leading to a delayed, reduced, or even a complete lack of rejection [53,54,67]. Based on preclinical studies that showed minimal apparent adverse rejection effects, pMSCs were approved for allogeneic cell therapy studies [23,53,68,69]. We suggest that the expression of CD276 on pMSCs could account for the improved tolerance to allogeneic and xenogeneic implanted pMSCs when compared to MSCs from other sources.

In the current study, we further explored the regulation of CD276 expression on pMSCs grown in a defined clinical-grade, GMP-compliant medium incorporating human plasma and platelet lysate in an attempt to explain why pMSCs are well tolerated upon homologous application in non-HLA-matched recipients and even upon injection in xenogenic animals serving as cell therapy models. 

## 2. Results

### 2.1. Expression of Typical MSC Markers in the Endometrial and Fetal Parts of the Placenta

We investigated the expression of the typical mesenchymal antigens in the cryosections of human-term placenta tissue samples from which pMSCs were isolated for this study. CD73, CD90, and CD105 expressions were observed in the fetal part of the chorion plate tissues, as well as in the villous-rich endometrial tissue of the placenta (Appendix A). In some areas, the CD73 was clearly detected on placenta syncytiotrophoblasts, which could not serve as a cell source for isolation and further expansion (Appendix A). The proteoglycan NG2, a marker for MSC-like pericytes [70] (Appendix A), and the adhesion molecule CD146 [18,20,26] were detected in cells from both maternal and fetal placenta tissues (Appendix A). In some samples, NG2 was rather low in the fetal part of the placenta, while a prominent expression was seen in the corresponding tissue samples of the maternal tissues from the same placenta. In other placentas, a prominent NG2 expression was observed in both the fetal and maternal placental tissues (Appendix A). The expression of CD146 was inconsistent within the samples investigated. In one of the placental samples, a robust expression of CD146 was seen only in the chorionic villi of the maternal placenta and not in the fetal side; meanwhile, in another sample, it was low in both parts. In other samples, a dispersed expression was seen in the fetal placenta, and a prominent expression of CD146 was observed in the intervillous space of the maternal placenta (Appendix A). By contrast, the expression of immune-checkpoint antigen CD276 (alias B7-H3) [54] was noted in all samples investigated (Figure 1). 

### 2.2. Expansion and Differentiation of pMSCs Derived from Maternal and Fetal Part of Term Placentas

Mesenchymal stromal cells were isolated from tissues dissected from the fetal-only and maternal-only parts of five human-term placentas and expanded as separate populations in either CME medium or GMP medium. All pMSC populations examined exhibited a characteristic fibroblast-like morphology (Figure 2). The pMSCs were characterized as mesenchymal stromal cells by the expression of typical cell surface markers (Appendix A) [12,16]. The number of fpMSCs expanded in the CME medium during the early passages was relatively low compared to the other three populations (Figure 2A–D). The fpMSCs that expanded in the CME medium from the primary culture over three consecutive passages proliferated significantly slower compared to mpMSCs in the CME medium (*p* < 0.047) and much slower than fpMSCs in the GMP medium (*p* ≈ 0.063; Figure 2E). Contrary to our findings with fpMSCs and mpMSCs in the GMP medium, there were no notable variations in the proliferation rates of mpMSCs cultured in the CME medium compared to mpMSCs expanded in the GMP medium (Figure 2E). 

The pMSCs are of relatively stable cell phenotypes and were therefore not expected to differentiate or to trans-differentiate to other phenotypes as reported from multipotent bmMSCs. However, to comply with the MSC inclusion criteria mentioned above [12], the ability of fpMSCs and mpMSCs to undergo adipogenic and osteogenic differentiation, when expanded in the CME medium or GMP medium, respectively, was evaluated. The phenotype of the adipocytes was determined by staining their intracellular lipid vesicles with Oil Red O (Figure 3). Unlike bmMSCs, the induced osteogenic differentiation of pMSCs was lower [17,18,19,26] (Figure 4). Morphological changes, such as flat-stretched cells and a minor calcification of the extracellular matrix were observed in cells expanded in the CME medium (Figure 4A,C). However, both the fpMSCs and mpMSCs that expanded in the GMP medium showed little or almost no in vitro osteogenesis (Figure 4B,D). Table 1 presents an overview of the evaluation of the in vitro differentiation of pMSCs. This verifies the stability of these cells, with only minimal potential for the induction of differentiation. The behavior of the pMSCs as adult differentiated mesenchymal cells with limited differentiation potential was also confirmed by a very low expression of the typical stem cell marker Sushi-contining-domain-2 (SUSD2) in both pMSC populations tested (Figure 5). 

### 2.3. Expression of Cell Surface Antigens on pMSCs Expanded in CME Versus GMP Medium

The expression of cell surface markers was investigated in early passage pMSCs using FC (Figure 5, Appendix A). The fpMSCs expanded in the CME medium expressed the typical consensus MSC surface markers CD73, CD90, and CD105 [12,48], with no expression of the hematopoietic antigens CD14, CD34, and CD45 (Appendix A). Most fpMSCs expressed high levels of both CD146 and CD276, while some also expressed SUSD2 at low to moderate levels (Appendix A). The fpMSCs from the same placenta expanded in the GMP medium had similar expression profiles (Appendix A). However, the expressions of CD105, CD146, and SUSD2 on fpMSCs, grown in the GMP medium, were lower. Moreover, some cells expressed somewhat more CD14 compared to the same population expanded in CME media (Appendix A). Both fpMSCs and mpMSCs isolated from the same placenta and expanded in the CME medium exhibited nearly identical antigen expression profiles (Appendix A). Similarly, both fpMSCs and mpMSCs from the same placenta expanded in the GMP medium and had similar antigen expression profiles (Appendix A). Overall, the expression of CD73 was prominent in all populations investigated, while the expression of CD105 was found to be lower (Figure 5). Similarly, a robust expression of CD90 was found in the above three sets of cell populations (Figure 5). A low expression of typical hematopoietic CD34 and CD45 was recorded in the pMSCs expanded in cells expanded in CME and GMP media. By contrast, the CME medium induced a significant difference in the expression of CD90 in mpMSCs (*p* < 0.007) compared to mpMSCs cultured in the GMP medium. The GMP medium elevated the expression of CD14 in fpMSCs compared to mpMSCs in the same medium (*p* < 0.05, Figure 5); however, the mean MFI of the expression of CD14 was clearly below the levels observed for the positive markers CD73 and CD90. The expression of CD146 was significantly lower in fpMSCs expanded in the CME medium (*p* < 0.05, Figure 5). The expression of the immune checkpoint antigen CD276 was significantly higher in mpMSCs in the GMP medium compared to the same population expanded in the CME medium (*p* < 0.001) and to fpMSCs expanded in the GMP medium (*p* < 0.01; Figure 5). Moreover, the histograms of SUSD2-stained cells appeared wider with higher MFI peak signal intensities in fpMSCs and mpMSCs expanded in the CME medium compared to the same cells expanded in the GMP medium (Appendix A). However, significant differences were not observed in the transcription levels.

We further investigated whether the significant changes in the CD146 and CD276 expressions recorded using FC were derived from changes in the regulation of the gene expression of these markers (Figure 6). Significant transcript levels encoding CD146 were found in both mpMSCs expanded in the GMP medium (*p* < 0.05) and CME medium (*p* < 0.001) compared to fpMSCs in the GMP medium (Figure 6A). This corroborated the FC results (Figure 5) and suggests the need to regulate CD146 expression on the transcript level. A different effect was recorded for CD276 transcript expression wherein mpMSCs (*p* < 0.04) and fpMSCs in the GMP medium (*p* < 0.02) expressed significantly higher CD276 expression compared to mpMSCs in the CME medium (Figure 6B). These accumulated data suggest that factors in the GMP medium significantly increased the expression of CD276 on the surface of mpMSCs. This effect seemed to be regulated, at least in part, by the steady-state transcription levels in these cells.

## 3. Discussion

Numerous studies have provided evidence that MSCs isolated from various tissues have rather low rejection rates and, therefore, can be utilized as allogeneic implants for relatively prolonged periods with the modulation of immune responses to exert their indirect pro-regenerative effect for an extended period before their clearance [33,49,51,71,72,73]. In this respect, both isolated fpMSCs and mpMSCs seem to be good candidates for allogeneic treatments with non-matched cells. Such tolerance seems to be mainly attributed to a low expression of HLA class II antigens [49,72,74] and a high expression of indoleamine 2,3-dioxygenase 1 [75]. The reduced rejection rate could also be attributed to the lack of CD80 (B7-1) and CD86 (B7-2) expression, the co-stimulatory ligands required for an efficient co-activation of T-lymphocytes by interacting with the CD28 or CTLA-4 receptors [76]. While HLA class II can be upregulated on MSCs by factors such as INFγ and TGF-β, CD80 and CD86 expression seemed to remain low upon proinflammatory stimulation [77]. Other costimulatory cell surface molecules include CD274 (PD-L1, B7-H1) and CD276 (B7-H3), which are both expressed in bmMSCs [78,79,80], and these must be taken into account when considering the MSC-induced immune-regulatory effects.

The expression of CD274 was reported to be higher in fetal placental wjMSCs than in bmMSCs, with no significant dependence on the composition of the media employed for cell expansions [81,82]. By contrast, the expression of CD276 was reported to be higher in bmMSCs compared to wjMSCs. Nevertheless, no comparison of its expression in pMSCs relative to other MSCs was given in the previous studies [81,82]. One could conclude, as we suggest now, that MSCs inherit their immune modulatory capacities from a combination of the very low expression of HLA class II, the low expression of costimulatory proteins CD80 and CD86, as well as the prominent expression of co-regulatory proteins CD274 and/or CD276. In general, a significant expression of CD276 was determined in maternal and fetal tissues of the human placenta, with the highest expression levels occurring in the pMSCs. This may explain the higher immune tolerance of the mother towards the allogeneic fetus that she carries [83]. 

The high expression of CD276 was found on transcript levels and on the cell surfaces of pMSCs following their in vitro expansion in the GMP medium. More than 97% of all pMSCs gated were found to be CD276-positive. The mean fluorescence intensities (MFI) of CD276 on cells expanded in the CME medium ranged from 3520 to 3960, while, in the GMP medium, the MFIs ranged from 4040 to 6250. The CD276 expression levels of pMSCs seemed to be even higher than the expression of typical MSC marker antigens CD105 and CD146 [26], and they were similar to the expression levels of the classical MSC markers CD73 and CD90 [12]. We propose that the platelet lysate utilized in the preparation of the GMP medium may enhance the expression of CD276. 

Our preliminary data suggest that mpMSCs with similar transcript levels of CD276 to fpMSC may present more efficiently on the cell surfaces. This suggests that the translation of the CD276 protein and/or its transport to the cell surface may be regulated differently in these distinct cells. Regulatory miRNAs may possibly take a role in this process; however, further studies are needed to confirm this point. It should be considered that, due to the low numbers of fpMSCs obtained in early passages in the CME cultures, the effect of the GMP medium on the regulation of CD276 could not be investigated in detail. More dedicated detailed studies should address this point. 

The specific effects on the individual ligands and receptors in the GMP medium are of interest. The expansion of pMSCs in the GMP medium did not induce a general upregulation or downregulation on any cell surface molecule. While the expression of CD90 was significantly higher in cells in the CME medium, the expression of CD73, CD105, and SUSD2 was not affected by the alteration of the medium. The expression of CD146 in MSCs was associated with an efficient osteogenic differentiation, thereby promoting undesired ectopic osteogenesis [18,19,26]. The significantly low expression of CD146 in fpMSCs in combination with the low risk of osteogenic differentiation could, therefore, represent a considerable advantage of these cells in different clinical applications. The distinct regulation of the expression of CD90 and CD146 compared to CD276 in pMSCs in the GMP medium emphasizes the significance of the use of adequate cell culture protocols for achieving the desired cells’ phenotype to be used for preclinical and later clinical studies.

To obtain the maximal effect of pMSC-based treatments, we expect the cells to have a stable prototype with minimal spontaneous and/or induced differentiation. Our study confirms the rather low osteogenic differentiation potential of both pMSC subsets compared to bmMSC [18,19]. SUSD2, which is downregulated by TGF-ß1, has been described as a marker for differentiation competence and associated with the “stemness” of multipotent stem cells, such as bmMSCs [45]. Low SUSD2 was also associated with cellular senescence [84]. Our data suggest a trend toward a lower expression of SUSD2 in pMSCs expanded in the GMP medium, which contains TGF-β1 contributed by the platelet lysate [85]. This seems to contrast with the faster proliferation of cells cultured in the GMP medium compared to the same cells in the CME medium. Further studies on cells expanded to high passages should address this important issue which may be relevant for the potential clinical applications of high numbers of pMSCs following their multi-passage expansions [45]. The stable phenotype of the pMSCs with limited differentiation potential (“stemness”) seems to render them proper candidates for allogeneic treatments, in which they could exert their pro-regenerative effect without undesired differentiation. The relatively slow proliferation rate of pMSCs in the CME medium makes their production longer while increasing the risks of the contamination of the culture. Therefore, the rapid expansion of CD276^high^ in pMSCs in the GMP medium for only a few passages seems preferable in the preparation of pMSC stocks to be used for clinical application [68].

The expression of the hematopoietic markers CD34 and CD45 was found to be low on all pMSCs and was not significantly affected by the medium. Nevertheless, an expression of CD14 was observed in fpMSCs expanded in the GMP medium using the monoclonal antibody clone M5E2. This does not necessarily indicate the presence of monocytes in the cultured fpMSCs. As typical monocytes are CD14^pos^, we expect them to become visible by FC as a minor peak or a shoulder aside from the main peak of the fpMSCs. But this was not observed. The positive CD14 staining, which was observed only on some batches of pMSCs, may be caused by the cross-reactivity with a relevant epitope, as previously reported [86]. In the clinical applications of such cells, if this issue turns out to be critical, minor numbers of cells expressing CD14, or of cells expressing a CD14 cross-reactive epitope, could be removed by cell sorting. 

The use of allogenic pMSCs could potentially become a clinical solution for MSC-based cell therapy, especially in conditions in which autologous cells are either not readily available in a timely manner or insufficient in numbers to address emergency situations, such as strokes or heart attacks. This approach is especially relevant when the autologous MSCs from elderly patients may yield less promising efficacies [38,40,68].

## 4. Materials and Methods

### 4.1. Collection of Tissues from the Human Placenta

The Ethics Committee of the University Hospital in Tubingen approved the use of the donated human placenta with pre-signed mothers’ informed consent (file number 364/2018B02). Five placentae in pre-scheduled Caesarian deliveries from healthy mothers and newborns were collected in sterile conditions in the operation room of the labor ward of the University Hospital in Tubingen. The placentae were thoroughly washed in a buffered physiological solution and were kept chilled for less than one hour before their further processing.

### 4.2. Preparation of Cryosections of Placenta Tissue Samples for Immunofluorescence Analysis

Small tissue samples were cut from the intact fetal or maternal tissues of the placenta, and cryosections were cut, (10 μm, Leica CM1860 UV, Leica Biosystems, Nußloch, Germany), mounted on slides, air-dried, and blocked (10% pre-immune serum of species of the detection antibody in PBS-Tween20, 30 min, 20 °C; Appendix A). Then, the sections were washed twice (PBS) and incubated with the primary antibody (Appendix A) in a humidified box (60 min, 20 °C). Primary antibodies were washed off (3×, 20 °C, PBS-Tween20), and the relevant fluorescently labeled detection antibody was added and further incubated (60 min, 20 °C, humidified dark box; Appendix A). The specific detection antibody was washed off and the slides were covered with a mounting solution containing DAPI (Vectashield, BIOZOL, Eching, Germany) to visualize cell nuclei and covered with coverslips. The sections were viewed and photographed using a fluorescence microscope (Axiovert 200 M, 20× objective; Zeiss, Jena, Germany). The micrographs were recorded and processed using proprietary software (Axiovision 4.8, Zeiss). Cryosections without primary antibodies and incubation with detection reagent only served as controls.

### 4.3. Expansion of pMSCs Isolated from Maternal and Fetal Tissues of Full-Term Placentas

The pMSCs were isolated from term placental tissues as previously described [17,18,19,20]. For the isolation of fpMSCs from the fetal part, the amnion membrane was removed, and the relevant tissue samples were dissected from the fetal-only tissue of the surface of the chorion. The mpMSCs were isolated from maternal tissue samples from the endometrial third of the placenta. The tissue samples from both sources were washed twice (HBSS; Sigma–Aldrich, Steinheim, Germany) and carefully minced into small fragments, rinsed, and collected via centrifugation. The sediments were resuspended in a digestion solution (750 U/mL collagenase (Sigma–Aldrich), 250 mg/mL Dispase II, (Roche, Basel, Switzerland) in HBSS), and further incubated (60 min at 37 °C). The proteolysis was blocked using a solution with 10% FCS, and the debris was removed through gradient centrifugation (Ficoll-Paque cushion, 1.077 g/L, 800× *g*, 20 min, 20 °C) and filtered with a 70–100 µm cell mesh. The interphase was collected and washed twice with PBS solution. For the expansion of the cells and for further analyses, the fpMSCs and mpMSCs were resuspended in two different expansion media. One medium was the clinical-grade, GMP-compliant expansion medium (GMP medium: low glucose DMEM, 1000 IE heparin, 2 mM L-glutamine, 2% penicillin-streptomycin, 1% amphotericin B, complemented with 5% pooled human plasma and 5% pooled human platelet extract) [20]. The other was a commercial MSC expansion medium (CME medium; MSCGM^®^ plus BulletKit; Lonza, Washington, DC, USA). Cell proliferation was enumerated according to the expansion of the cells for three consecutive passages and cell counting was performed with the aid of a hematocytometer and trypan dye exclusion; the duplication rate per day was enumerated. 

### 4.4. Differentiation of pMSCs In Vitro

Induction of adipogenic and osteogenic differentiation was performed by incubating the pMSCs in differentiation media for 3 weeks [17]. The adipogenic differentiation was induced in a DMEM high-glucose medium complemented by 10% FBS, insulin, indomethacin, 3-isobutylxanthin, dexamethasone, Pen/Strep, and amphotericin-B. The adipogenic differentiation was verified by staining the cells with Oil Red O. The osteogenic differentiation was induced using an osteogenic medium consisting of a DMEM high-glucose medium, 10% FBS, dexamethasone, β-glycerophosphate, ascorbic acid with Pen/Strep, and amphotericin B. The mineralization of the cell cultures and osteogenic differentiation were visualized with von Kossa staining. As controls to all differentiation experiments, cells were seeded in a GMP starvation medium, based on DMEM low-glucose medium, 0.1% plasma, and 0.1% platelet extract, heparin, HEPEs, glutamine, and Pen/Strep.

### 4.5. Flow Cytometry (FC) Analyses of Cell Surface Markers on pMSCs

For FC analyses, pMSCs were cultured and expanded to the second passage in either the CME or GMP medium. The cells were washed, harvested using mild proteolysis, washed again with PBS, and counted. Then, 5 × 10^5^ cells were incubated with the primary antibodies (Appendix A) diluted in PFEA buffer (PBS, 2% FCS, 2 mM EDTA, 0.01% sodium azide, 4 °C) complemented by 5% pre-immune globulins (Gammunex, Grifols, 60 min, 4 °C in the dark) [17]. The antibodies were washed off twice with PFEA buffer and the cells were resuspended in 200 μL PFEA buffer, diluted with FACS-buffer (BD BioSciences, San Jose, CA, USA), and analyzed using FC (LSR II, BD BioSciences) and proprietary software (FACS Diva V6.1.3 and FloJo V10.7, BD BioSciences) according to the manufacturer’s instructions [87]. For each experiment, compensation with unstained cells and with Comb Beads (BD BioSciences) was performed. Dead cells and debris were excluded according to supplier protocol by gating the cells for subsequent fluorescence analyses using the forward scatter (FSC) and side scatter (SSC).

### 4.6. Determination of Steady-State Transcript Levels using PCR of cDNA Produced from mRNA

For the analysis of the gene expression reflected in the transcript levels, the pMSCs were expanded to the second passage in either the CME or GMP medium, washed, and detached from the culture flasks [17]. RNA was then extracted (RNeasy mini kit; Qiagen, Venlo, The Netherlands), and cDNA was synthesized using reverse transcription (PrimeScript 1st strand cDNA kit; TaKaRa, San Jose, CA, USA). For quantitative PCR analyses, cDNA was mixed with a master mix (Roche) and specific primers (Appendix A). The target cDNAs were amplified (35 cycles: 94 °C 30″, 60 °C 30″, 72 °C 20″), complemented by product extension (72 °C 300″), and cooled to 4 °C (LightCycler480; Roche). GAPDH and PPIAγ amplifications served as housekeeping controls in each experiment. In each run, the expression of CD276 was normalized to the PCR products of GAPDH and PPIAγ with the δδCt algorithm (Roche LC480 manual for real-time PCR (RT-qPCR) [88].

### 4.7. Statistics

A statistical evaluation of the significance of the experimental results was performed using two-sided, unbiased *t*-tests along with the spreadsheet program Excel for Mac (Microsoft, V16.78.3; Redmond WA, USA), where *p*-values of <0.05 were considered significant. 

## 5. Conclusions

The results of the current exploratory study indicate that the expression of the immune checkpoint antigen CD276, alias B7-H3, is high in pMSCs expanded in the GMP-compliant cell expansion medium. The elevated expression of CD276 may contribute to the very low immunogenicity of pMSCs upon allogeneic application in the clinical studies reported [68] by mitigating cytotoxic T-lymphocyte responses. It may also contribute to immune tolerance reported from the xenogeneic application of human pMSCs in animal models [23]. The immune tolerance towards human pMSCs injected in non-primate species facilitates the functional studies of the regenerative potential of pMSC-based cell therapies in various small and large animal models. Future studies will investigate if the application of pMSCs supports a functional regeneration of the sphincter complex in a model of stress urinary incontinence.

## Figures and Tables

**Figure 1 ijms-24-16422-f001:**
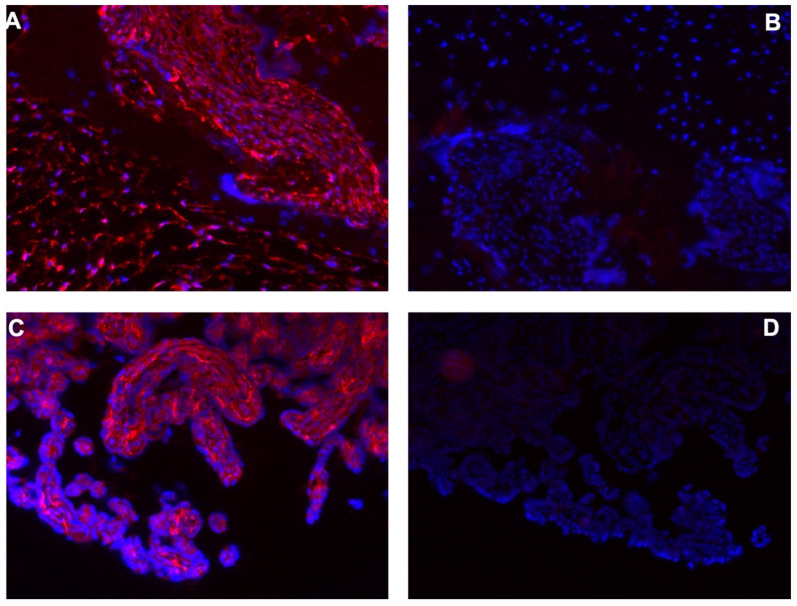
Detection of CD276 in human-term placenta. Cryosections were generated from the fetal part (**A**,**B**) and maternal part of the placenta (**C**,**D**). The cyosections were stained using an anti-CD276 antibody followed by a Cy-3-labelled detection antibody (**A**,**C**), or using a detection antibody only (**B**,**D**). Samples were counterstained using DAPI to visualize the nuclei. All micrographs were taken with a 20× objective. CD276-positive cells were detected in both the fetal and the maternal tissue samples taken from the term placenta.

**Figure 2 ijms-24-16422-f002:**
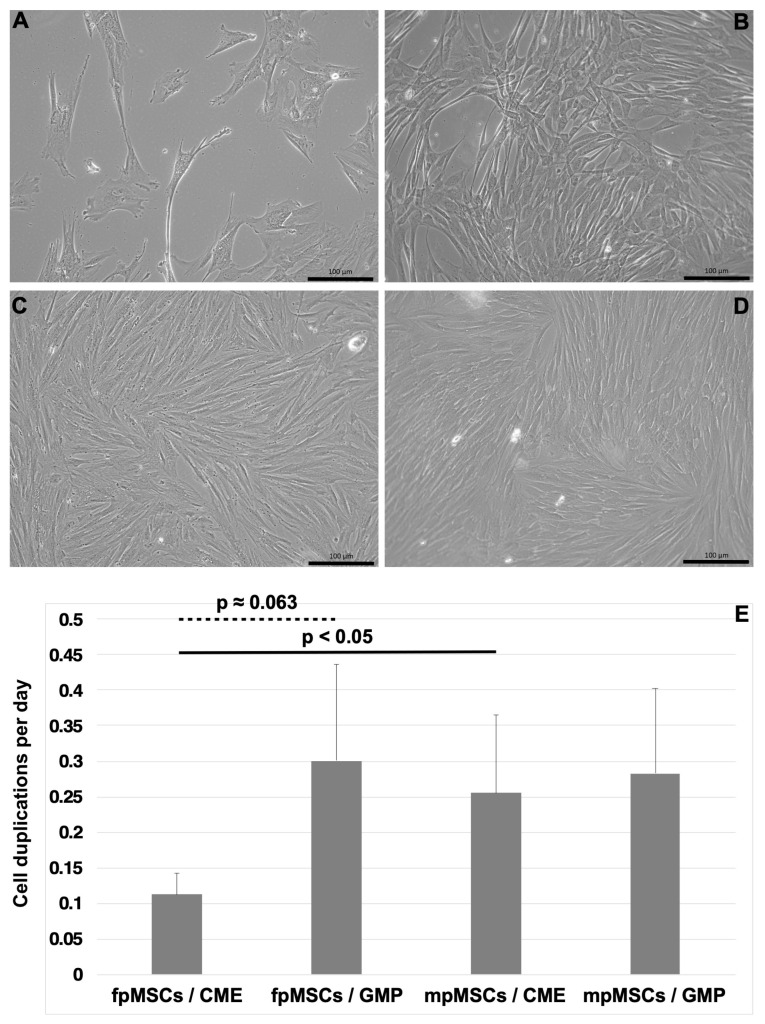
Expansion of pMSCs in vitro in different media. Placenta-derived MSCs in their first passage of in vitro culture derived from tissue samples from the fetal (**A**,**B**) and maternal parts (**C**,**D**) of the same placenta were expanded ex vivo in CME media (**A**,**C**) or GMP media (**B**,**D**). The pMSCs grow adherent as fibroblast-like cells. The size bars indicate 100 μm (**A**–**D**). The cell proliferation was enumerated over three consecutive passages (**E**). A significant difference was noted between the fpMSCs and mpMSCs expanded in CME media (solid line). A trend of the accelerated proliferation of mpMSCs in the CME media, compared to the slower proliferation of fpMSCs in the GMP media, was noted (dashed line).

**Figure 3 ijms-24-16422-f003:**
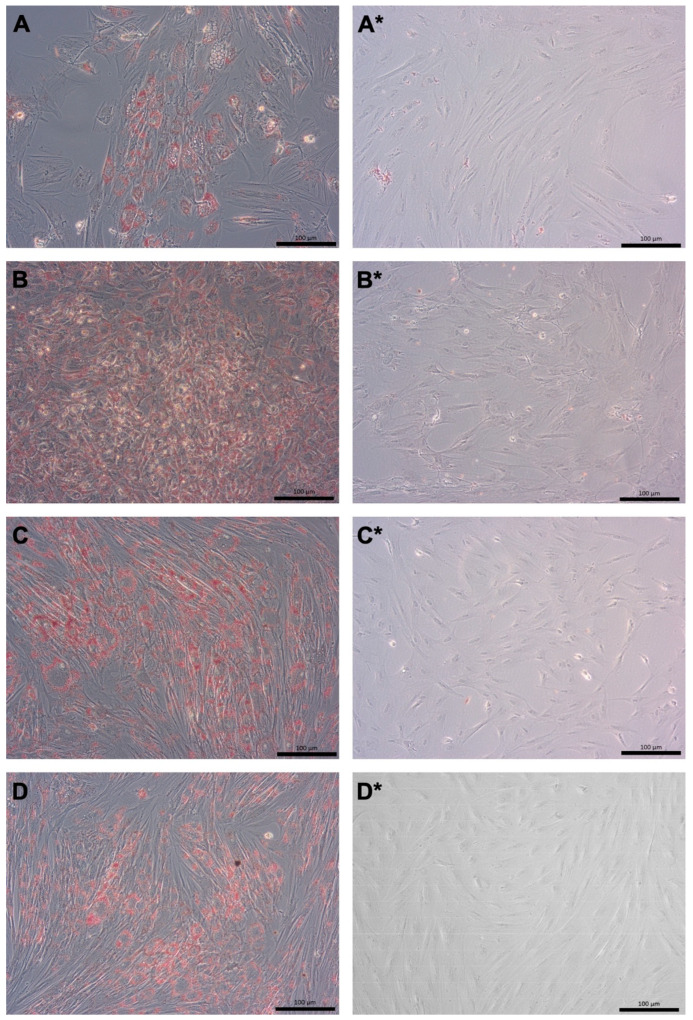
Adipogenic differentiation of pMSCs in vitro. The pMSCs from the fetal part (**A**–**B***) and the maternal part (**C**–**D***) from the same placenta were expanded in either CME (**A**,**A***,**C**,**C***) or GMP (**B**,**B***,**D**,**D***) media, and adipogenic differentiation was induced for three weeks (**A**–**D**). pMSCs expanded in starvation medium served as controls (**A***–**D***). Adipogenesis was determined by staining the lipophilic vesicles in the cells with Oil Red O. Lipid vesicles were found in all of these cell populations following induction of adipogenesis, confirming the differentiation capacity in fpMSCs and mpMSCs expanded in both CME and GMP media. Size bars indicate 100 μm.

**Figure 4 ijms-24-16422-f004:**
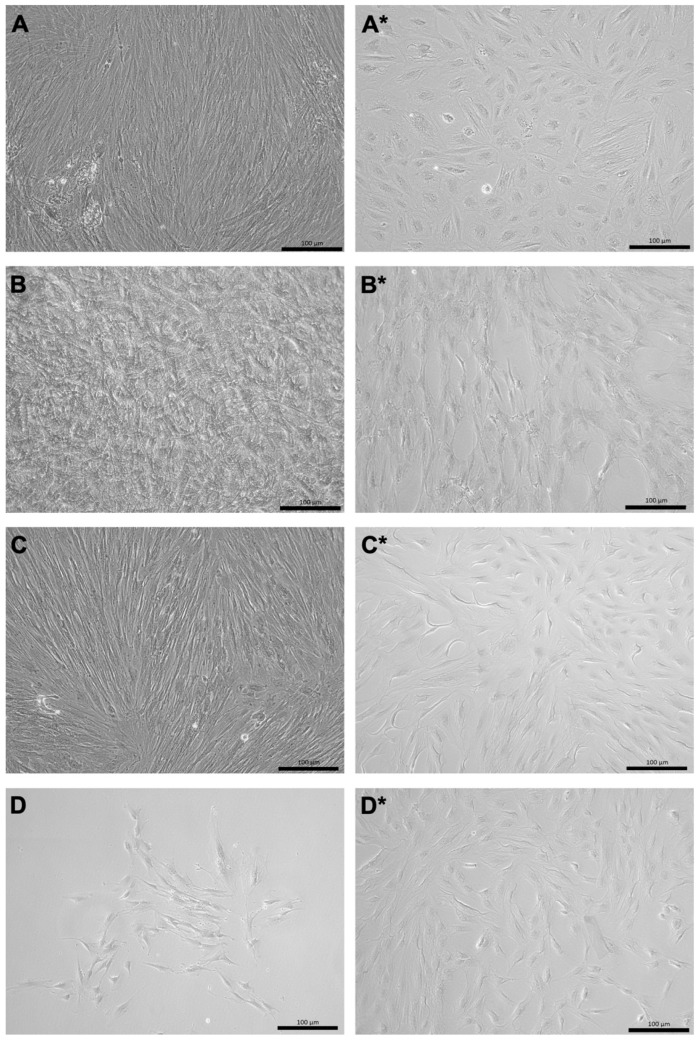
Osteogenic differentiation of pMSCs in vitro. The pMSCs from the fetal part (**A**–**B***) and the maternal part (**C**–**D***) from the same placenta were expanded in either CME (**A**,**A***,**C**,**C***) or GMP (**B**,**B***,**D**,**D***) media. Osteogenic differentiation was induced for three weeks (**A**–**D**). pMSCs expanded in a starvation medium and served as negative controls (**A***–**D***). Progress of osteogenesis was determined by staining the mineralized matrix components with von Kossa staining. The cultures expanded in CME medium prior to osteogenic differentiation appear somewhat darker (**A**,**C**) when compared to the cultures expanded in GMP medium (**B**,**D**). Nevertheless, areas of a mineralized extracellular matrix of the cultures were not detected in any of the cell populations investigated, confirming that osteogenic differentiation is not expected to be a prominent process in pMSC cultures. Size bars indicate 100 μm.

**Figure 5 ijms-24-16422-f005:**
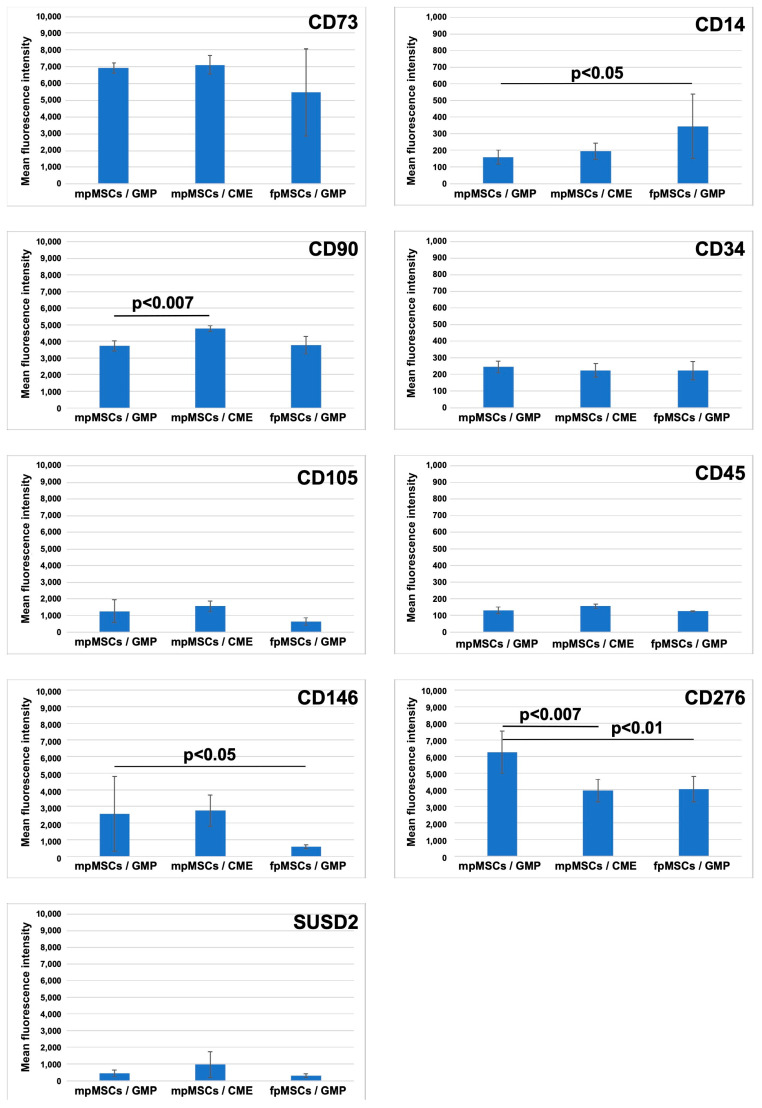
Analysis of expression of cell surface markers on pMSCs using FC. The fpMSCs and mpMSCs were expanded in both GMP or CME medium to the second passage of in vitro culture as indicated in each diagram (X-axis). The expression of the positive MSCS markers CD73, CD90, CD105, the exclusion markers CD14, CD34, CD45, the osteogenic marker CD146, the immune checkpoint antigen CD276, and the MSC stem cell marker SUSD2 were assayed using FC. The mean MFI ± S.D. from analyses of pMSCs from individual placentas is presented on the Y-axis in each diagram. The pMSCs expressed the marker proteins in typical patterns. Significant regulation of CD90 (*p* < 0.007), CD146 (*p* < 0.05), and CD276 (*p* < 0.01 and 0.007, respectively) were noted as indicated. As fpMSCs proliferated slowly in CME medium, a statistical evaluation of cells from at least 3 individual donors was impossible with such cells. However, an example of a complete set of analyses, obtained using FC, is presented in Appendix A. Note the difference in MFI scales for CD14, CD34, and CD45 adopted to disclose even minor changes in staining intensities.

**Figure 6 ijms-24-16422-f006:**
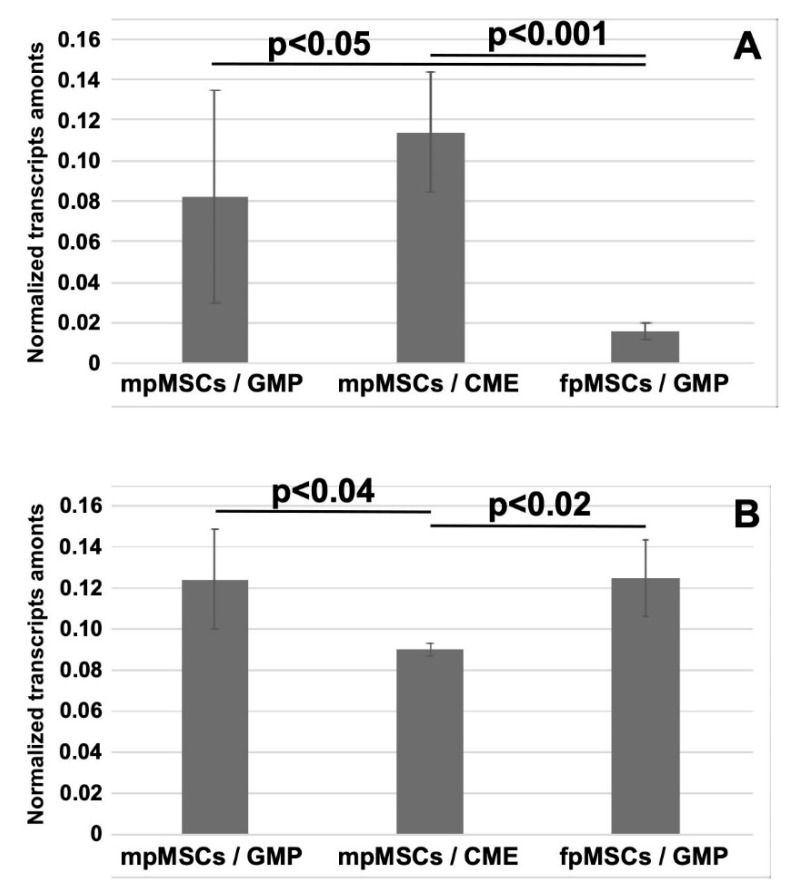
Analysis of expression of cell surface markers on pMSCs using RT-qPCR. Cells from the maternal and fetal parts of the human-term placenta were expanded in a GMP or CME medium to the second passage of in vitro culture as indicated in the diagram. RNA was extracted, cDNA generated, and the transcripts encoding CD146 (**A**) and CD276 (**B**) were enumerated in each population and normalized to GAPDH and PPIAγ as housekeeping genes. The normalized gene expression index ± standard deviations is shown on the Y-axis, and the cell types and culture conditions are shown on the X-axis. Transcripts encoding the osteogenic marker CD146 are significantly lower in fpMSCs expanded in the GMP medium compared to mpMSCs in the same medium (*p* < 0.05) or mpMSCs in the CME medium (*p* < 0.001; (**A**)). Expression of CD276 is significantly elevated in mpMSCs (*p* < 0.04) and fpMSCs in the GMP medium (*p* < 0.02) compared to fpMSCs in the CME medium (**B**).

**Table 1 ijms-24-16422-t001:** Overview of differentiation efficacies of pMSCs depending on the expansion media. The pMSCs from fetal and maternal zones were expanded in either CME or GMP media to their second passage of the in vitro culture. Adipogenic or osteogenic differentiation was induced. Cell incubation in starvation media served as a control. The overall differentiation efficacy is indicated by an arbitrary scale using “+” for weak to “+++++” for very strong, while the failure of differentiation is disclosed by “ø”.

Cells	Medium	Oil Red O	Adipogenesis	Von Kossa	Osteogenesis
fpMSCs	CME	2/3 weak	+	2/3 weak	+
		1/3 medium	+++	1/3 medium	++
fpMSCs	GMP	1/5 weak	+	1/weak	+
		3/5 medium	+++	3/5 little	++
		1/5 strong	++++	1/5 medium	+++
mpMSCs	CME	1/5 weak	+	2/5 weak	+
		4/5 medium	+++	3/5 medium	+++
mpMSCs	GMP	1/5 weak	+	ø	ø
		3/5 medium	+++	ø	ø
		1/5 very strong	+++++	ø	ø

## Data Availability

The raw databases of the presented information are available to academic colleagues for non-commercial purposes upon their justified request.

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
