# Peer review of "Regulation of Immune Checkpoint Antigen CD276 (B7-H3) on Human Placenta-Derived Mesenchymal Stromal Cells in GMP-Compliant Cell Culture Media"

_ijms, 2023, doi:10.3390/ijms242216422_

Round 1

Reviewer 1 Report

Comments and Suggestions for Authors

The aim of the study by Amend et al. was to further elucidate the known effect that hPMSCs are well tolerated despite HLA mismatching. They cultured hPMSCs in two different media and compared antigen expression by flow cytometry, immunofluorescence, and qRT-PCR.

Overall, the manuscript is well written, and in most parts comprehensible. The structure of the manuscript appears unorganized in some parts (which might be due to converting problems). The study design is clear, yet further elucidating experiments are missing and the results are not always strong enough for the conclusions drawn.

There are many concerns with this report:

1)      Structure

a)       Table 1 and figure 2 are positioned in the middle of the methods section but present regular results

b)      Table 1 is not formatted (due to converting from word document to pdf?), and was very hard to read

c)       Figure 2 is the first figure in the manuscript, figure 1 the second

d)      Abbreviations are inconsistent: CEM medium (table 1 and legend of table 1) vs. CME medium (all other figures and text)

e)      f-hPMSC vs. fpMSC, same for m-hPMSC and mpMSC (please standardize abbreviations)

2)      Methods, legends and figure display

a)       Figure 1: why is Alexa Fluor 488 staining depicted in red? In the online supplement figures AF488 staining is depicted in green (as commonly used). Also, according to online supplement table 1 the primary antibody is rabbit anti human CD276, the secondary antibody should hence be the noted (anti rabbit) Cy3.

b)      Similar question for os figure 5, why is AF488 staining depicted in red, contrary to os figures 1-4?

c)       How was PMSC proliferation measured or “enumerated”? The term “relative proliferation index” is not described. Statistical significance is stated without standard deviations in the diagram. Also, in line 282, the authors state, that f-hPMSCs proliferated “slightly” slower in CME-medium compared to GMP-medium. Yet, the difference is the biggest among the four cell/medium combinations. The “slightly” slower proliferation then results in too few cells for further analyses.

d)      Proliferation analysis would benefit from standardized viability/proliferation assays like MTT/MTS

e)      Figure 6: y-axis “Normalized transcript amonts” is lacking numbers. How many fold increased? Does it translate to biological significance (MFI)? Also: were technical triplicates analyzed or biological quintuplets?

3)      Data interpretation and discussion

a)       The main result of the study was that GMP medium induced “a significant expression of CD276” in mpMSCs compared to CME medium. The MFI is indeed 50% higher compared to mpMSC in CME medium. In figure 6, CD276 transcript levels corroborate the findings. On the other hand, MFI levels of fpMSC are significantly lower compared to mpMSCs/GMP, but CD276 transcript levels are equal. The authors don`t discuss this discrepancy. For comparison: expression and transcript levels for CD146 are coherent for all cell/medium combinations. The authors then suggest that CD146 expression is regulated on transcript levels (line 381), but “a different effect was recorded for CD276”. Yet, the effect is the same for mpMSC, and for fpMSC the data on fpMSC/CME are missing, hence a conclusion can’t be drawn. MSC from other origins should be employed to clarify the impact of GMP-medium on CD276 expression.

Furthermore: CD146 expression was reported to be rather heterogeneous between the five donors, resulting in high standard deviations (figures 5 and 6). Were the surface expression and transcript level data coherent for each donor?

b)      CD276 is also expressed on bmMSC (line 404), and expressed higher compared to wjMSC. Therefore, to truly describe the impact of CD276 expression on immune tolerance towards allogeneic PMSC, the expression of CD276 on both/other cell origins needs to be compared side by side, best in different media.

c)       The title claims that CD276 contributes to the reduced immune response against (all?) human placenta-derived mesenchymal stromal cells. This is misleading, if the cell culture conditions have an impact on CD276 expression. How does this finding translate to in vivo application of these PMSC? Is the effect reversible if the GMP media cultured PMSCs are transferred to CME media?

Is the expression of CD276 upregulated by GMP medium or downregulated by CME medium?

d)      The authors address the necessity of high passages for clinical application, yet they present no data on GMP vs CME medium expansion beyond passage 3.

e)      CD14 expression on fpMSCs with clone M5E2 should be clarified with a staining with clone MΦP9 and not simply “interpreted” as crossreactivity.

f)        Which factor in GMP medium is responsible for increase in CD276 expression? Please include in discussion or even better perform experiments.

Author Response

We thank the reviewers for their excellent and detailed suggestions which will all help to significantly improve the quality of the study as well as the quality of its presentation. Attached please find a point-by-point replay. Our responses are written in italics below each point.

Reviewer 1:

The aim of the study by Amend et al. was to further elucidate the known effect that hPMSCs are well tolerated despite HLA mismatching. They cultured hPMSCs in two different media and compared antigen expression by flow cytometry, immunofluorescence, and qRT-PCR. 

Overall, the manuscript is well written, and in most parts comprehensible. The structure of the manuscript appears unorganized in some parts (which might be due to converting problems). The study design is clear, yet further elucidating experiments are missing and the results are not always strong enough for the conclusions drawn.

There are many concerns with this report:

1)      Structure

  1. a) Table 1 and Figure 2 are positioned in the middle of the methods section but present regular results

Ad 1a) This point is possibly a misconception derived from pointing to online supplement table 1, originally laid out as “os Table 1”. The indication to the real “Table 1” was and is in the results section and deals with the differentiation of PMSCs. We therefore rephrase the indications to the online tales to “osTable 1” etc.

  1. b) Table 1 is not formatted (due to converting from word document to pdf?), and was very hard to read

Ad 1b) We apologize for this inconvenience and reformatted Table 1 using the “format table menu” of the software employed. We hope that this revised table will not be skewed by changing the document to a PDF or alike.

  1. c) Figure 2 is the first figure in the manuscript, figure 1 the second

Ad 1c) During revision we checked the course of figures mentioned in the text. The “rea figure 1” is metioned in chapter 1 of the Results, the “real figure 2) in the second chapter of the results. We, therefore, think – unless we missed something -  should stick to the order of figures.

  1. d) Abbreviations are inconsistent: CEM medium (table 1 and legend of table 1) vs. CME medium (all other figures and text)

Ad 1d) We do apologize for this misspelling. Of course, in Table 1 and its legend CEM must be replaced by CME. This was revised accordingly.

  1. e) f-hPMSC vs. fpMSC, same for m-hPMSC and mpMSC (please standardize abbreviations)

Ad 1e) Again, we thank this reviewer for this critique and revised the abbreviations in the text to match the labels in Figures 2, 5, and 6, respectively. We hope that we did not write in new misspellings during our revision!

2)      Methods, legends and figure display

  1. a)Figure 1: why is Alexa Fluor 488 staining depicted in red? In the online supplement figures AF488 staining is depicted in green (as commonly used). Also, according to online supplement table 1 the primary antibody is rabbit anti human CD276, the secondary antibody should hence be the noted (anti rabbit) Cy3. 

Ad 2a) The reviewer is quite correct! As can be seen from osTable 1, figure 1 presents the expression of CD276 which is detected by a rabbit IgG antibody and detected by Cy3-labeled detection antibody. We revised the legend accordingly.

  1. b)Similar question for os figure 5, why is AF488 staining depicted in red, contrary to os figures 1-4?.

Ad 2b) Good catch! Detection of CD146 was facilitated by a PE-labeled antibody. We revised the legend to this figure accordingly.

  1. c)How was PMSC proliferation measured or “enumerated”? The term “relative proliferation index” is not described. Statistical significance is stated without standard deviations in the diagram. Also, in line 282, the authors state, that f-hPMSCs proliferated “slightly” slower in CME-medium compared to GMP-medium. Yet, the difference is the biggest among the four cell/medium combinations. The “slightly” slower proliferation then results in too few cells for further analyses.

Ad 2c: The cells were counted by routine methods using a hematocytometer and trypan blue dye exclusion. This method was added during the revision to the M%&M section. The term “relative proliferation index” was replaced during the revision and the more common term “cell duplication rate” is used in the revised version of the manuscript. In Figure2E of the revision the atandard deviations are included. As the difference of the proliferation of fPMSCs in CME was not significantly different when compared to fPMSCs in GMP, we replaced the phrase “slightly” by the word “clearly” in the revised manuscript.

  1. d)Proliferation analysis would benefit from standardized viability/proliferation assays like MTT/MTS

Ad 2d. We do agree with the reviewer in part. One advantage of cell counting by a hematocytometer and trypan dye exclusion is that it requires only a few cells. We still recorded significant differences among the populations investigated. Proliferation assays are advantageous in a variety of experiments. For this study, however, we have to go along with the data we do have at hand.

  1. e)Figure 6: y-axis “Normalized transcript amonts” is lacking numbers. How many fold increased? Does it translate to biological significance (MFI)? Also: were technical triplicates analyzed or biological quintuplets? 

Ad 2e) The numbers on the y-axis of Figure 6 were unfortunately “hiding” under the label of the y-axis. By revision of Figure 6, the numbers were added to indicate the differences in the normalized transcript levels.

3)      Data interpretation and discussion

  1. a)The main result of the study was that GMP medium induced “a significant expression of CD276” in mpMSCs compared to CME medium. The MFI is indeed 50% higher compared to mpMSC in CME medium. In figure 6, CD276 transcript levels corroborate the findings. On the other hand, MFI levels of fpMSC are significantly lower compared to mpMSCs/GMP, but CD276 transcript levels are equal. The authors don`t discuss this discrepancy. For comparison: expression and transcript levels for CD146 are coherent for all cell/medium combinations. The authors then suggest that CD146 expression is regulated on transcript levels (line 381), but “a different effect was recorded for CD276”. Yet, the effect is the same for mpMSC, and for fpMSC the data on fpMSC/CME are missing, hence a conclusion can’t be drawn. MSC from other origins should be employed to clarify the impact of GMP-medium on CD276 expression.

Furthermore: CD146 expression was reported to be rather heterogeneous between the five donors, resulting in high standard deviations (figures 5 and 6). Were the surface expression and transcript level data coherent for each donor?

Ad3a) We do agree with this important point and in the revised discussion this discrepancy was briefly addressed. However, a detailed discussion was impossible as reviewer 2 requested a condensation of the document. We therefore have to balance the addition of information against the extension of the text.

Referring to osFig. 5, the expression of CD146 was heterogeneous in tissue samples as mentioned in the text. The standard deviations of CD146 expression as recorded in pMSCs upon expansion by FC (Fig. 5) and RT-qPCR (Fig. 6) are variable as well. However, the focus of the study was to investigate the regulation and/or expression of CD276. We therefore refrain from overinterpretation of our data in the context of CD146. As this is an interesting question, it should be explored in a future study.

  1. b)CD276 is also expressed on bmMSC (line 404), and expressed higher compared to wjMSC. Therefore, to truly describe the impact of CD276 expression on immune tolerance towards allogeneic PMSC, the expression of CD276 on both/other cell origins needs to be compared side by side, best in different media. 

Ad3b) We agree that CD276 is expressed on MSCs prepared from different tissues. However, this study is focused on the regulation of CD276 on pMSCs, including cells from the maternal versus fetal parts. Therefore, a complete study of the regulation of CD276 on any or many other types of MSCs from the very different tissues that harbor such cells goes clearly beyond the scope of this study.

  1. c)The title claims that CD276 contributes to the reduced immune response against (all?) human placenta-derived mesenchymal stromal cells. This is misleading, if the cell culture conditions have an impact on CD276 expression. How does this finding translate to in vivoapplication of these PMSC? Is the effect reversible if the GMP media cultured PMSCs are transferred to CME media?

Is the expression of CD276 upregulated by GMP medium or downregulated by CME medium?

Ad 3c) In the context of many other studies, the title seems not misleading. Indeed, CD276high cells are known to provoke less immune reactions when compared to the CD276low cells. But this was not studied in detail in the current experiments. We therefore revised the title. Moreover, the experiment was not designed to study the mechanism of up- or down-regulation on molecular levels in detail. This would, e.g., include analyses of steady-state mRNA concentrations in different media, in timelines (kinetics), promoter-reporter analyses, and others. The term “elevated” does not mean in the context of our study an activation of the corresponding promotor but only that one finds more RNA under a given condition when compared to other conditions, and by no means a mechanism of regulation of gene expression. However, we complied with this issue by moderately revising the wording of the corresponding paragraphs.

  1. d)The authors address the necessity of high passages for clinical application, yet they present no data on GMP vs CME medium expansion beyond passage 3. 

Ad3d) Actually, the placenta is such a rich source for MSCs that extended expansion of cells is probably not required for many applications. The possibility of extended expansion is an option and the comparably “young placenta ells” are better suited for extended expansion when compared to the problems of replicative senescence of MSCs upon in vitro expansion derived from elderly donors.

  1. e)CD14 expression on fpMSCs with clone M5E2 should be clarified with a staining with clone MΦP9 and not simply “interpreted” as crossreactivity.

Ad3e) We do we simply “interpret” the data of CD14 reactive fpMSCs expanded in GMP medium as crossreactivity as we state (quote) “an expression of CD14 was observed”. So crossreactivity is only one interpretation and we provide a citation in support. However, we do not deny that some of the cells in this population may have been CD14-positive. Anyway,  adapted this paragraph to better point to this critical issue of CD14 reactive cells.

  1. f)Which factor in GMP medium is responsible for increase in CD276 expression? Please include in discussion or even better perform experiments.

Ad3f) We clearly state in the manuscript that factors contained in the GMP media are the best candidates responsible for maintaining an elevated or for inducing an elevated expression of CD276 on pMSCs. Among others, bFGF could contribute to this effect. As GMP media contains many factors, the experiments suggested would require an enormous effort in resources and time. For the advancement of science such studies seem adequate, but, as pointed out above, such experiments are clearly beyond the focus of this more clinical study investigating the role of CD276 on pMSCs produced under GMP-compliant conditions for possible allogenic treatment of patients, including the elderly suffering e.g. from an emergency situation which cannot be managed by time-consuming production of autologous cells for therapy.

Reviewer 2 Report

Comments and Suggestions for Authors

The original article “Contribution of checkpoint antigen CD276 (B7-H3) to the reduced immune response against human placenta-derived mesenchymal stromal cells” reported that CD276 expression on hPMSCs was increased in culture using GMP medium. CD276 expression can prevent immunological attack host immune cells when transplantation is done. This topic is very interesting because immune-check point molecules might be a key to success for several organ transplantation and cell therapy. I considered that this article was suitable for publication for “Internal Journal of Molecular Sciences”, but several issues was present before acceptance.

1.       Introduction and methods session was too long to read. These parts should be reduced.

2.       hPMSCs in GMP medium showed high CD276 expression by FC and PCR. When this method was used in clinical practice, it can be important that CD276 expression maintains after medium is changed from GMP to another one, such as human primary serum. The author should do the additional investigation about that.  

Author Response

We thank the reviewers for their excellent and detailed suggestions which will all help to significantly improve the quality of the study as well as the quality of its presentation. Attached please find a point-by-point replay. Our responses are written in italics below each point.

Reviewer 2:

The original article “Contribution of checkpoint antigen CD276 (B7-H3) to the reduced immune response against human placenta-derived mesenchymal stromal cells” reported that CD276 expression on hPMSCs was increased in culture using GMP medium. CD276 expression can prevent immunological attack host immune cells when transplantation is done. This topic is very interesting because immune-check point molecules might be a key to success for several organ transplantation and cell therapy. I considered that this article was suitable for publication for “Internal Journal of Molecular Sciences”, but several issues was present before acceptance.

  1. Introduction and methods session was too long to read. These parts should be reduced.
  2. hPMSCs in GMP medium showed high CD276 expression by FC and PCR. When this method was used in clinical practice, it can be important that CD276 expression maintains after medium is changed from GMP to another one, such as human primary serum. The author should do the additional investigation about that. 

We thank the reviewer for the critiques and advised the manuscript accordingly:

Ad1: Reviewer 1 requested in-depth discussion and more details in the M&M parts ( e.g. cell counting)  But we agree with reviewer 2 that the introduction was in part redundant and not structured well. The same applies to the M&M. We therefore revised these two chapters. The revised introduction was cut by about 12% text, M&M by 11%, despite the additions requested.

Ad2: Our GMP medium contains platelet lysate and human plasma. In our previous studies, this blend was determined as a well-suited medium for GMP-compliant production of MSCs. However, it is clear that a change of the cell culture medium and for instance the replacement of platelet lysate by human serum requires new experiments. Additional studies along these lines would require systematic experiments with fpMSCs and mpMSCS from several deliveries, rt-qPCR, FC, immunofluorescence, etc., a task simply not possible within the focus of this study, nor as complementary data to be added here. But we appreciate this hint and consider that in future studies depending on the clinical need.

Round 2

Reviewer 1 Report

Comments and Suggestions for Authors

Reviewer 1:

The aim of the study by Amend et al. was to further elucidate the known effect that hPMSCs are well tolerated despite HLA mismatching. They cultured hPMSCs in two different media and compared antigen expression by flow cytometry, immunofluorescence, and qRT-PCR. 

Overall, the manuscript is well written, and in most parts comprehensible. The structure of the manuscript appears unorganized in some parts (which might be due to converting problems). The study design is clear, yet further elucidating experiments are missing and the results are not always strong enough for the conclusions drawn.

There are many concerns with this report:

1)      Structure

  1. a) Table 1 and Figure 2 are positioned in the middle of the methods section but present regular results

Ad 1a) This point is possibly a misconception derived from pointing to online supplement table 1, originally laid out as “os Table 1”. The indication to the real “Table 1” was and is in the results section and deals with the differentiation of PMSCs. We therefore rephrase the indications to the online tales to “osTable 1” etc.

-       Table 1 still is positioned in the middle of the methods sections (line 138), but the other figures are in the right order in the results section now.

  1. b) Table 1 is not formatted (due to converting from word document to pdf?), and was very hard to read

Ad 1b) We apologize for this inconvenience and reformatted Table 1 using the “format table menu” of the software employed. We hope that this revised table will not be skewed by changing the document to a PDF or alike.

-       ok

  1. c) Figure 2 is the first figure in the manuscript, figure 1 the second

Ad 1c) During revision we checked the course of figures mentioned in the text. The “rea figure 1” is metioned in chapter 1 of the Results, the “real figure 2) in the second chapter of the results. We, therefore, think – unless we missed something -  should stick to the order of figures.#

-       see above

  1. d) Abbreviations are inconsistent: CEM medium (table 1 and legend of table 1) vs. CME medium (all other figures and text)

Ad 1d) We do apologize for this misspelling. Of course, in Table 1 and its legend CEM must be replaced by CME. This was revised accordingly.

-       There are still mix ups regarding CEM and CME throughout the text (compare lines 256, 305, 316, 349, 354, 444, legend to fig. 4)

-       Other interesting typos:

o    Line 221/222 material placenta

o    Line 224 H7-H3

o    Line 239 paternal tissue of the placenta???

o    Line 265 CD14 twice

o    Line 344 0<0.02

  1. e) f-hPMSC vs. fpMSC, same for m-hPMSC and mpMSC (please standardize abbreviations)

Ad 1e) Again, we thank this reviewer for this critique and revised the abbreviations in the text to match the labels in Figures 2, 5, and 6, respectively. We hope that we did not write in new misspellings during our revision!

-       ok

2)      Methods, legends and figure display

  1. a)Figure 1: why is Alexa Fluor 488 staining depicted in red? In the online supplement figures AF488 staining is depicted in green (as commonly used). Also, according to online supplement table 1 the primary antibody is rabbit anti human CD276, the secondary antibody should hence be the noted (anti rabbit) Cy3. 

Ad 2a) The reviewer is quite correct! As can be seen from osTable 1, figure 1 presents the expression of CD276 which is detected by a rabbit IgG antibody and detected by Cy3-labeled detection antibody. We revised the legend accordingly.

-       ok

  1. b)Similar question for os figure 5, why is AF488 staining depicted in red, contrary to os figures 1-4?.

Ad 2b) Good catch! Detection of CD146 was facilitated by a PE-labeled antibody. We revised the legend to this figure accordingly.

-       ok

  1. c)How was PMSC proliferation measured or “enumerated”? The term “relative proliferation index” is not described. Statistical significance is stated without standard deviations in the diagram. Also, in line 282, the authors state, that f-hPMSCs proliferated “slightly” slower in CME-medium compared to GMP-medium. Yet, the difference is the biggest among the four cell/medium combinations. The “slightly” slower proliferation then results in too few cells for further analyses.

Ad 2c: The cells were counted by routine methods using a hematocytometer and trypan blue dye exclusion. This method was added during the revision to the M%&M section. The term “relative proliferation index” was replaced during the revision and the more common term “cell duplication rate” is used in the revised version of the manuscript. In Figure2E of the revision the atandard deviations are included. As the difference of the proliferation of fPMSCs in CME was not significantly different when compared to fPMSCs in GMP, we replaced the phrase “slightly” by the word “clearly” in the revised manuscript.

-       ok

  1. d)Proliferation analysis would benefit from standardized viability/proliferation assays like MTT/MTS

Ad 2d. We do agree with the reviewer in part. One advantage of cell counting by a hematocytometer and trypan dye exclusion is that it requires only a few cells. We still recorded significant differences among the populations investigated. Proliferation assays are advantageous in a variety of experiments. For this study, however, we have to go along with the data we do have at hand.

-       ok

  1. e)Figure 6: y-axis “Normalized transcript amonts” is lacking numbers. How many fold increased? Does it translate to biological significance (MFI)? Also: were technical triplicates analyzed or biological quintuplets? 

Ad 2e) The numbers on the y-axis of Figure 6 were unfortunately “hiding” under the label of the y-axis. By revision of Figure 6, the numbers were added to indicate the differences in the normalized transcript levels.

-       ok

3)      Data interpretation and discussion

  1. a)The main result of the study was that GMP medium induced “a significant expression of CD276” in mpMSCs compared to CME medium. The MFI is indeed 50% higher compared to mpMSC in CME medium. In figure 6, CD276 transcript levels corroborate the findings. On the other hand, MFI levels of fpMSC are significantly lower compared to mpMSCs/GMP, but CD276 transcript levels are equal. The authors don`t discuss this discrepancy. For comparison: expression and transcript levels for CD146 are coherent for all cell/medium combinations. The authors then suggest that CD146 expression is regulated on transcript levels (line 381), but “a different effect was recorded for CD276”. Yet, the effect is the same for mpMSC, and for fpMSC the data on fpMSC/CME are missing, hence a conclusion can’t be drawn. MSC from other origins should be employed to clarify the impact of GMP-medium on CD276 expression.

Furthermore: CD146 expression was reported to be rather heterogeneous between the five donors, resulting in high standard deviations (figures 5 and 6). Were the surface expression and transcript level data coherent for each donor?

Ad3a) We do agree with this important point and in the revised discussion this discrepancy was briefly addressed. However, a detailed discussion was impossible as reviewer 2 requested a condensation of the document. We therefore have to balance the addition of information against the extension of the text.

Referring to osFig. 5, the expression of CD146 was heterogeneous in tissue samples as mentioned in the text. The standard deviations of CD146 expression as recorded in pMSCs upon expansion by FC (Fig. 5) and RT-qPCR (Fig. 6) are variable as well. However, the focus of the study was to investigate the regulation and/or expression of CD276. We therefore refrain from overinterpretation of our data in the context of CD146. As this is an interesting question, it should be explored in a future study.

-       The question on data coherence pointed at the statement that CD146 expression was regulated on transcript level, which could be nicely demonstrated by the heterogeneous individual donor’s data.

  1. b)CD276 is also expressed on bmMSC (line 404), and expressed higher compared to wjMSC. Therefore, to truly describe the impact of CD276 expression on immune tolerance towards allogeneic PMSC, the expression of CD276 on both/other cell origins needs to be compared side by side, best in different media. 

Ad3b) We agree that CD276 is expressed on MSCs prepared from different tissues. However, this study is focused on the regulation of CD276 on pMSCs, including cells from the maternal versus fetal parts. Therefore, a complete study of the regulation of CD276 on any or many other types of MSCs from the very different tissues that harbor such cells goes clearly beyond the scope of this study.

-       Ok, since the authors clearly stated the limitations of the study.

  1. c)The title claims that CD276 contributes to the reduced immune response against (all?) human placenta-derived mesenchymal stromal cells. This is misleading, if the cell culture conditions have an impact on CD276 expression. How does this finding translate to in vivoapplication of these PMSC? Is the effect reversible if the GMP media cultured PMSCs are transferred to CME media?

Is the expression of CD276 upregulated by GMP medium or downregulated by CME medium?

Ad 3c) In the context of many other studies, the title seems not misleading. Indeed, CD276high cells are known to provoke less immune reactions when compared to the CD276low cells. But this was not studied in detail in the current experiments. We therefore revised the title. Moreover, the experiment was not designed to study the mechanism of up- or down-regulation on molecular levels in detail. This would, e.g., include analyses of steady-state mRNA concentrations in different media, in timelines (kinetics), promoter-reporter analyses, and others. The term “elevated” does not mean in the context of our study an activation of the corresponding promotor but only that one finds more RNA under a given condition when compared to other conditions, and by no means a mechanism of regulation of gene expression. However, we complied with this issue by moderately revising the wording of the corresponding paragraphs.

-       Ok

  1. d)The authors address the necessity of high passages for clinical application, yet they present no data on GMP vs CME medium expansion beyond passage 3. 

Ad3d) Actually, the placenta is such a rich source for MSCs that extended expansion of cells is probably not required for many applications. The possibility of extended expansion is an option and the comparably “young placenta ells” are better suited for extended expansion when compared to the problems of replicative senescence of MSCs upon in vitro expansion derived from elderly donors.

-       Why not include this fact in the discussion right after “Further studies on cells expanded to high passages should address this important issue which may be relevant for potential clinical applications of high pMSCS numbers following their multi-passage expansions.”?

  1. e)CD14 expression on fpMSCs with clone M5E2 should be clarified with a staining with clone MΦP9 and not simply “interpreted” as crossreactivity.

Ad3e) We do we simply “interpret” the data of CD14 reactive fpMSCs expanded in GMP medium as crossreactivity as we state (quote) “an expression of CD14 was observed”. So crossreactivity is only one interpretation and we provide a citation in support. However, we do not deny that some of the cells in this population may have been CD14-positive. Anyway,  adapted this paragraph to better point to this critical issue of CD14 reactive cells.

-       ok

  1. f)Which factor in GMP medium is responsible for increase in CD276 expression? Please include in discussion or even better perform experiments.

Ad3f) We clearly state in the manuscript that factors contained in the GMP media are the best candidates responsible for maintaining an elevated or for inducing an elevated expression of CD276 on pMSCs. Among others, bFGF could contribute to this effect. As GMP media contains many factors, the experiments suggested would require an enormous effort in resources and time. For the advancement of science such studies seem adequate, but, as pointed out above, such experiments are clearly beyond the focus of this more clinical study investigating the role of CD276 on pMSCs produced under GMP-compliant conditions for possible allogenic treatment of patients, including the elderly suffering e.g. from an emergency situation which cannot be managed by time-consuming production of autologous cells for therapy.

-       Agreed, that experiments are out of the scope of this study. Yet, a bit more in-depth discussion on possible regulating factors would be desirable (but also agreed: not necessary at this point)

Author Response

Dear Editor, Dear Reviewer 1,

Again, we are very thankful for the excellent comments and are kind of embarrassed that several misspellings remained in R1, especially several CEMs, were still found by this reviewer. We now checked and revised  these errors to the best of our possibilities including e.g., the search as well ass search-replace functions of MS-Word software. We hope to present now a manuscript that merits publication in the IJMS. Below please find a second point-by-point reply:

Ad 1a) in reference to online supplement Table 1: we wrote in version R1:

 “….  pre-immune serum of species of the detection antibody in PBS-Tween20, 30 min. 20°C; online supplement (os) Table 1). Then the sections were washed twice (PBS) and incubated with the primary antibody (osTable 1) …“ .

In the document R2-clear, line 142, we rephased these lines to adapt the text to the request of the reviewer 1 (quote):

“ …. pre-immune serum of species of the detection antibody in PBS-Tween20, 30 min. 20°C; online supplement Table 1 (osTab. 1)). Then the sections were washed twice (PBS) and incubated with the primary antibody (osTab. 1), …”

We do hope that this further change now matches the request of reviewer 1.

Ad 1d: during the second revision, CEM was changed in CME throughout the text.

Line 253: material placenta was changed in maternal placenta

Line 256: H7-H3 was changed in B7-H3

Line 259. paternal was revised in maternal

Lines 546: one CD14 was revised to be CD34

Line 563: 0<0.02 was revised in p < 0.02